# Risk of Heart Failure between Different Metabolic States of Health and Weight: A Meta-Analysis of Cohort Studies

**DOI:** 10.3390/nu14245223

**Published:** 2022-12-08

**Authors:** Xiaowen Wang, Jiayi Dong, Zhicheng Du, Jie Jiang, Yonghua Hu, Liqiang Qin, Yuantao Hao

**Affiliations:** 1Center for Public Health and Epidemic Preparedness & Response, Peking University, Beijing 100191, China; 2Public Health, Department of Social Medicine, Osaka University Graduate School of Medicine, Osaka 5650871, Japan; 3Department of Epidemiology and Biostatistics, School of Public Health, Peking University Health Science Center, Beijing 100191, China; 4Department of Medical Statistics, School of Public Health, Sun Yat-sen University, Guangzhou 510275, China; 5Department of Nutrition and Food Hygiene, School of Public Health, Soochow University, Suzhou 215000, China

**Keywords:** metabolic health, obesity, heart failure, cohort, meta-analysis

## Abstract

We conducted a systematic review of cohort studies comparing the risk of heart failure in people with differing metabolic health and obesity statuses. We searched three electronic databases (PubMed, Web of Science, Scopus), where the studies of the relationships of metabolic health and obesity statuses with heart failure were included. Fixed-effects or random-effects models were used to estimate the summary relative risks [RRs]. Ten cohort studies were selected. Compared with individuals with normal metabolic health and body mass, the pooled RRs (95% confidence intervals) for heart failure were 1.23 (1.17, 1.29) for metabolic healthy overweight individuals, 1.52 (1.40, 1.64) for metabolic healthy individuals with obesity, 1.56 (1.30, 1.87) for metabolically unhealthy normal-weight individuals, 1.75 (1.55, 1.98) for metabolically unhealthy overweight individuals, and 2.28 (1.96, 2.66) for metabolic unhealthy individuals with obesity. A sensitivity analysis suggested that no single study had a substantial effect on the results. The Egger’s and Begg’s tests showed no evidence of publication bias. People with overweight or obesity were at a higher risk of heart failure, even if metabolically healthy. In addition, compared with metabolically healthy normal-weight individuals; metabolically unhealthy normal-weight individuals, and those with overweight or and obesity, were at higher risk of heart failure.

## 1. Introduction

Heart failure has become a global public health concern, with an estimated 64.34 million people being affected [1]. Heart failure is associated with high morbidity and mortality, implying a large clinical burden and necessitating high healthcare expenditure [2,3]. Obesity is often accompanied by metabolic disorders, including diabetes, hypertension, and dyslipidemia, which play important roles in the pathogenesis and development of heart failure [4,5,6,7,8]. There is also a subset of individuals with obesity, but without metabolic disorders, who are referred to as having “metabolically healthy obesity” (MHO). However, recently, MHO has received much clinical and public health attention, because it is not considered to be a benign condition [9,10]. Individuals with MHO may be at a higher risk of developing cardiometabolic diseases than healthy lean people [11,12], and the study of MHO may also provide new perspectives regarding the mechanisms linking obesity to metabolic and cardiovascular abnormalities [11,13].

The relationships of obesity status and metabolic health status with the risk of heart failure have not been thoroughly evaluated. A cohort study by Voulgari et al. published in 2011 was the first on this topic. They reported that compared metabolically healthy normal-weight individuals, metabolically unhealthy normal-weight individuals are at a higher risk of heart failure, whereas MHO individuals are at a non-significantly lower risk [14]. However, two other cohort studies showed that the risk of heart failure is higher in people with obesity, regardless of their metabolic health status [15,16]. A meta-analysis published in 2019 summarized the results of these two studies and found that MHO or metabolically healthy overweight status is not associated with the risk of heart failure [17]. Since then, the results of several cohort studies have been published, but the results have been inconsistent. Other studies have shown that compared with metabolically healthy individuals without obesity, MHO is associated with a higher risk of heart failure [18,19,20,21,22]. However, a recent cohort study showed that compared with metabolically healthy normal-weight individuals, there is a higher risk of heart failure in metabolically unhealthy normal-weight individuals and metabolically unhealthy individuals with overweight or obesity, but not in those with MHO [23].

Given the accumulating evidence and inconsistency of the results published to date, we aimed to conduct a systematic review and meta-analysis to compare the risks of heart failure among people with differing metabolic health and obesity statuses.

## 2. Materials and Methods

### 2.1. Literature Search

We performed a meta-analysis that was designed, performed, and reported according to the Meta-analyses Of Observational Studies in Epidemiology (MOOSE) guidelines [24]. We searched three electronic databases (PubMed, Scopus, and Web of Science) for eligible studies in which the relationships of metabolic health status and obesity status with the risk of heart failure were evaluated up until August 31, 2022. The keywords were selected, with the use of appropriate Boolean operators, for the literature strategy: “metabolic health” OR “metabolically healthy” OR “metabolic syndrome” AND (“obesity” OR “overweight”) AND “heart failure” AND (“cohort study” OR “prospective study”) AND “follow-up study”. We further filtered the article types by “review”, “systematic review” or “meta-analysis”. Language and time restrictions were not applied. We only searched full-text publications; abstracts and unpublished manuscripts were not considered.

### 2.2. Study Selection

A study was determined as eligible for inclusion in the meta-analysis when the following criteria were all met: (1) exposure defined using metabolic health status and obesity status, (2) outcomes including heart failure, (3) cohort design, and (4) risk estimates with 95% confidence intervals (CIs) for the relationship between the defined exposure and outcome were reported. The duplicate studies presented as the published articles titles, authors, and publication year were deleted, by using Endnote software, version 9, Clarivate, Philadelphia, US. In this process, three authors (X.W., J.Y.D., and Z.D.) worked independently to screen each record. After screening, the final study selection was completed by evaluating the full text of the selected ones, and any disagreements were resolved by discussion.

### 2.3. Data Extraction

X.W. and J.Y.D. reviewed the selected articles independently and discussed to generate a standard form to record the following data from each included study to describe their characteristics: the family name of the first author, year of publication, characteristics of the participants (age, sex, and location), sample size, study design, number of cases, duration of follow-up, assessments of metabolic health and obesity, assessments of heart failure, risk estimates with 95% CIs, and the results of the fully adjusted statistical model.

### 2.4. Quality Assessment

Study quality was evaluated using the Newcastle-Ottawa Scale, which includes assessments of study sample selection, group comparability, and the assessment of exposures and outcomes. Each study could be awarded a maximum of 9 points, with ≥7 points indicating high quality and <7 points indicating low quality. Two authors (X.W. and J.Y.D.) independently conducted the quality assessment.

### 2.5. Statistical Analysis

We defined six groups according to the metabolic health status and obesity status of the participants: a metabolically healthy normal-weight group, a metabolically healthy overweight group, a metabolically healthy obesity group, a metabolically unhealthy normal-weight group, a metabolically unhealthy overweight group, and a metabolically unhealthy obesity group. Three studies combined individuals with overweight or obesity; we regarded those in the two Asian studies [19,20] as having overweight (mean body mass index (BMI) < 30 kg/m^2^) and those in the single US study [23] as having obesity (mean BMI > 30 kg/m^2^). The metabolically healthy normal-weight group was treated as the reference group for the analysis. Relative risks (RRs) and their 95% CIs were used as the common measures of the relationships, and hazard ratios or incidence rate ratios were converted to RRs. The heterogeneity of the RRs across the studies was tested using Cochran’s Q statistic and *p* < 0.10, and we also calculated the *I*^2^ statistic to estimate the degree of inconsistency across the studies [25]. According to the results of heterogeneity testing, a fixed-effects or random-effects model was used to estimate each summary RR [26]. Subgroup analyses were conducted according to the study location, study sample size, and the duration of follow up to identify possible sources of heterogeneity and their modifying effects. A sensitivity analysis was also performed to evaluate the effect of single studies on the overall risk estimate by omitting one study at a time and analyzing the remaining data [27]. We used Egger’s test and Begg’s test to assess potential publication bias [27]. Statistical analyses were performed using STATA version 12.0 (StataCorp, College Station, TX, USA).

## 3. Results

We initially obtained 591 records from the three databases. The majority of the records were excluded by scanning the title and abstract, mainly because the research question was not relevant or because they were duplicates. A total of 13 studies were selected for full-text review. Of these, we excluded two studies [28,29] that did not report a relationship with heart failure and one study [21] because the data were from the same source as another. Thus, data from 10 studies were included in the meta-analysis (Figure 1).

The characteristics of the included studies are shown in Table 1. The 10 studies were published between 2011 and 2021, with five conducted in European countries, three in Asian countries, and two in the United States. Of the 10 studies, nine had a prospective cohort design and one had a retrospective cohort design. One study was conducted in women and the others were conducted in both men and women. The median duration of follow-up in these studies ranged from 3.1 to 17.0 years. A total of 435,446 cases of heart failure cases among 8,092,751 participants were identified. The definition of metabolic health was based on the number of abnormal metabolic conditions (i.e., hypertension, diabetes, and hyperlipidemia), which varied among the studies. Overweight and obesity were defined using BMI or waist circumference. All the studies used medical records for their outcome assessments. The covariates adjusted for in multivariable models principally comprised age, sex, smoking status, alcohol consumption status, and physical activity status (Appendix A).

Figure 2 shows the results of individual studies and the pooled RRs of heart failure for the metabolically healthy overweight/obesity groups vs. the metabolically healthy normal-weight group. No significant evidence of heterogeneity was identified for the metabolically healthy overweight group (*p* for heterogeneity > 0.10), and a fixed-effects model was used to generate the results. In contrast, a random-effects model was used for the metabolically healthy obesity group (*p* for heterogeneity < 0.001). Compared with the metabolic healthy normal-weight group, the pooled RRs of heart failure were 1.23 (1.17, 1.29) for the metabolic healthy overweight group and 1.52 (1.40, 1.64) for the metabolically healthy obesity group.

Figure 3 shows the results of the individual studies and the pooled RRs of heart failure for the metabolically unhealthy normal-weight, overweight, and obesity groups compared with the metabolically healthy normal-weight group. For these three comparisons, the results varied across the studies (all *p* for heterogeneity < 0.001), and a random-effects model was used for the pooled analysis. Compared with the metabolically healthy normal-weight group, the pooled RRs of heart failure were 1.56 (1.30, 1.87) for the metabolically unhealthy normal-weight group, 1.75 (1.55, 1.98) for the metabolically unhealthy overweight group, and 2.28 (1.96, 2.66) for the metabolic unhealthy obesity group.

Table 2 shows the results stratified according to study area (Western or Eastern countries). Because 5 of the 10 studies were conducted in Western countries, the results for western countries were very similar to those obtained from the overall analyses. The associations for the metabolically unhealthy normal-weight and overweight groups were weaker in Eastern countries. However, the association for the metabolically unhealthy obesity group was stronger in Eastern countries (only one Chinese study was included). The results stratified by sample size and duration of follow-up are presented in Appendix A. Overall, study sample size did not appear to modify the associations, while associations appeared to be stronger in studies with a shorter duration of follow-up (median < 10 years). Although level of heterogeneity decreased in several subgroups, it remained high in the analyses of metabolically unhealthy groups, particularly metabolically unhealthy obesity.

We next performed a sensitivity analysis by omitting one study in turn and combining the remaining data, and no single study was found to have a substantial effect on the overall results for any exposure (Appendix A). In addition, the results of Egger’s and Begg’s tests showed no evidence of publication bias (all *p* for publication bias > 0.26).

## 4. Discussion

In the present meta-analysis of 8,092,751 individuals, we evaluated the relationships of metabolic health status and obesity status with the risk of heart failure. We found that people with overweight and obesity had approximately 23% and 52% higher risks of developing heart failure, even if they were metabolically healthy. Compared with the metabolically healthy normal-weight group, the metabolically unhealthy normal-weight group was associated with an approximately 56% higher risk of heart failure, and this risk increased to 75% and 128% for the metabolically unhealthy overweight and obesity groups.

There is growing interest in the relationship between MHO and cardiovascular disease (CVD). Despite the absence of metabolic abnormalities in individuals with obesity, early changes may already be present that contribute to a higher risk of occurrence of CVD. MHO was found to be associated with abnormalities in several markers of subclinical CVD, including impaired vasoreactivity, high carotid artery intima-media thickness and aortic pulse wave velocity, high left ventricular mass and impaired function, and higher prevalences of coronary and aortic calcification [32,33]. Previous cohort studies have shown positive associations of MHO with hypertension [34], myocardial infarction, ischemic heart disease [35], and stroke [36]. The findings of the present study are consistent with “healthy” obesity not being benign with respect to cardiovascular health, thereby adding to the evidence that MHO is associated with a higher risk of heart failure, because individuals with overweight or obesity were found to be at higher risk of heart failure, regardless of metabolic status.

The potential mechanisms involved in this increase in risk of heart failure include hemodynamic changes and changes in cardiac anatomy. Adipose tissue may affect the circulating blood volume, resulting in increases in cardiac output and ventricular stroke volume, thereby predisposing toward left ventricular enlargement, hypertrophy, and concentric remodeling, all of which are linked to the development of heart failure [37,38]. Thus, individuals with MHO might have worse subclinical left ventricular or systolic function [39]. When obesity induces metabolic abnormalities such as hypertension, hyperlipidemia, high fasting glucose concentration, and insulin resistance, the phenotypes of metabolically unhealthy overweight or obesity may lead to a further decline in cardiac function and the progression of heart failure [40].

We found that metabolically unhealthy individuals with normal weight were still at a higher risk of heart failure, and this group had a similar estimated risk to the MHO group (approximately 50%) in the present study. Previous prospective cohort studies have shown that the risk of CVD in metabolically unhealthy normal-weight individuals are at even higher risk than those with MHO [41,42,43]. The severity of obesity was assessed using BMI, which may not reflect body fat distribution. For example, the Women’s Health Initiative Study recently showed that both a large amount of trunk fat and small amounts of gluteofemoral or leg fat were associated with higher risks of CVD in postmenopausal women with normal BMIs [44]. Moreover, genetic studies have shown that differing metabolic and weight phenotypes are characterized by variability in genes and the genetic effects on adipocyte differentiation, lipogenesis, and lipolysis [45]. Metabolically unhealthy normal-weight individuals are more likely to have low leg fat mass and cardiorespiratory fitness, but greater fat deposition in the liver, visceral obesity, impairment in insulin secretion, insulin resistance, and carotid intima-media thickness than metabolically unhealthy individuals with obesity [45].

Metabolic health status and obesity status may change during a study. In most studies, a single baseline measurement of exposure factors was made, and we found only two studies [20,31] that evaluated the relationships of the changes in metabolic health status and obesity status with heart failure. For example, Kim et al. reported that a transition from the metabolically healthy normal-weight phenotype to the metabolically unhealthy phenotype increased the risk of incident heart failure, with hazard ratios of 1.33 (1.15, 1.54) for normal weight and 1.67 (1.21, 2.32) for obesity. Participants with metabolically unhealthy obesity lost weight (transition to the metabolically unhealthy normal-weight phenotype) or experienced a restoration of metabolic health (transition to MHO) were still associated with a higher risk of heart failure, with hazard ratios of 1.76 (1.49, 2.07) and 1.65 (1.38, 1.97), respectively. These findings are also consistent with the risk of heart failure being even greater in those who acquire metabolic abnormalities and/or maintain their obesity (e.g., transition to the metabolically unhealthy normal-weight phenotype or MHO). Thus, further longitudinal studies with repeated measures should be performed to assess the importance of changes in obesity status and metabolic health in a real-world clinical setting, because these would have important implications for the importance of weight management and metabolic health maintenance with respect to the risk of heart failure.

Several limitations of this study should be acknowledged. First, heterogeneity is a common limitation of meta-analyses. In the present study, considerable heterogeneity was identified in the analyses of the metabolically unhealthy groups. Only the studies by Caleyachetty et al. [18] and Fauchier et al. [22] showed that sex may modify the associations of metabolic health and obesity with the risk of heart failure. For instance, when comparing metabolically unhealthy normal-weight individuals to metabolically healthy normal-weight individuals, the RRs were 1.53 (1.42, 1.65) for men and 1.90 (1.75, 2.06) for women, suggesting that sex may be a possible source of the heterogeneity [18]. However, sex-specific analyses were not performed in the other cohort studies, and therefore we were unable to perform separate meta-analyses for men and women. Other factors, such as age, socioeconomic status, or different lifestyles could also explain the heterogeneity. It is of great importance to conduct future studies to explore this association for individuals with certain characteristics. Second, the inclusion criteria bias is inevitable. For example, the definition of metabolic health was based on the numbers of metabolic disorders identified in the constituent studies, which may have been another source of heterogeneity. Third, the included studies typically defined obesity using BMI, and only one study [23] used waist circumference, which has been shown to be a better means of assessing obesity. Lastly, publication bias is a threat to the validity of any meta-analysis; however, Begg’s and Egger’s tests showed little evidence of such bias in the present study.

In conclusion, the present meta-analysis of previous cohort studies showed that people with overweight or obesity are at a higher risk of heart failure, even if they are metabolically healthy. Specifically, compared with metabolically healthy normal-weight individuals, metabolically unhealthy normal-weight individuals and those with overweight or obesity were found to be at a higher risk of heart failure. Therefore, the promotion of both metabolic health and weight management may represent an important means of reducing the incidence of heart failure.

## Figures and Tables

**Figure 1 nutrients-14-05223-f001:**
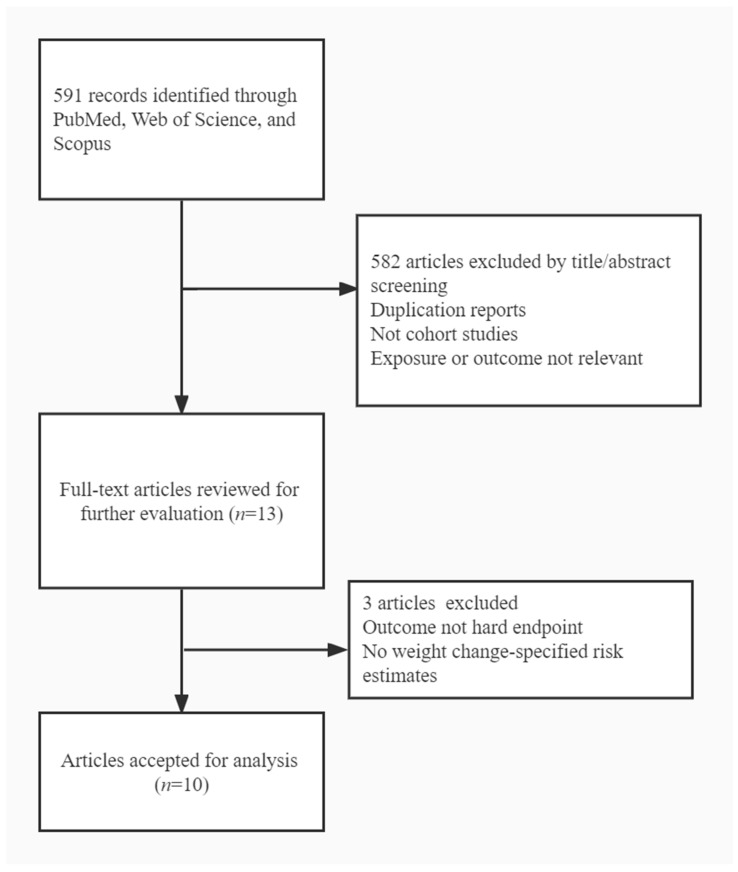
Flow chart of study selection.

**Figure 2 nutrients-14-05223-f002:**
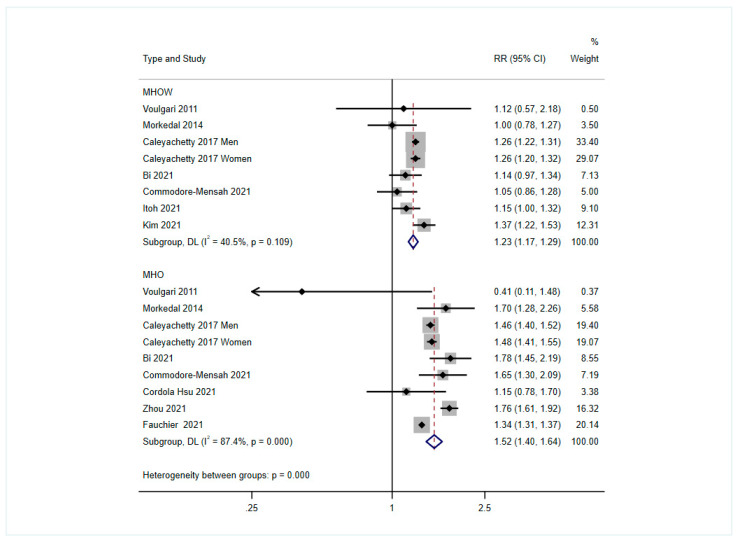
Pooled results of the comparison of the risk of heart failure associated with metabolically healthy overweight or obesity, compared with that of the metabolically healthy normal-weight group [14,15,16,18,19,20,22,23,30,31]. MHOW: Metabolically healthy overweight group, MHO: Metabolically healthy obesity group.

**Figure 3 nutrients-14-05223-f003:**
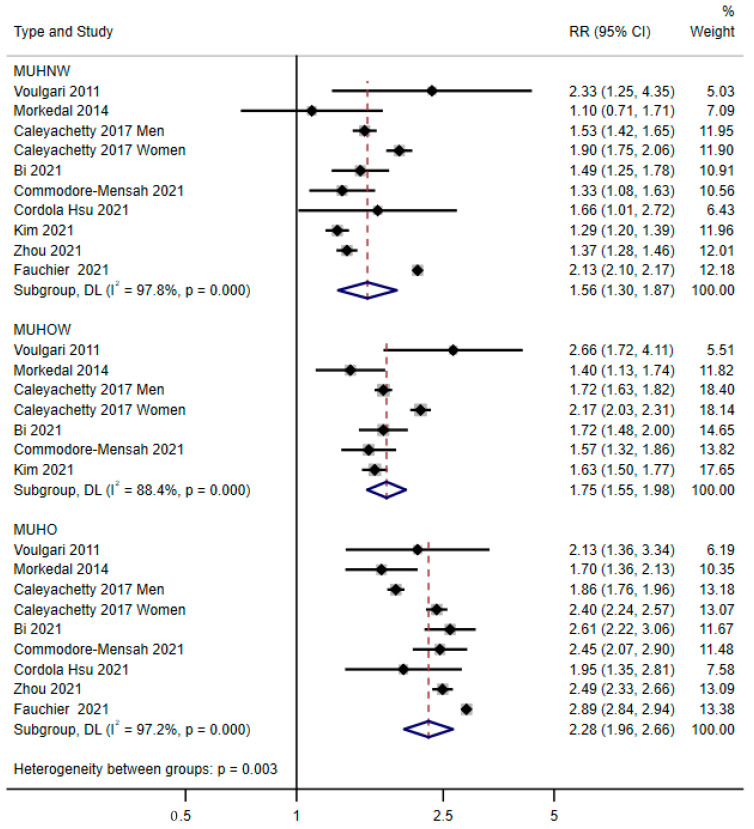
Pooled results of the comparison of the risk of heart failure associated with metabolically unhealthy normal weight, overweight, or obesity, compared with that of the metabolically healthy normal-weight group [14,15,16,18,19,20,22,23,30,31]. MUHNW: Metabolically unhealthy normal-weight group, MUHOW: Metabolically unhealthy overweight group, MUHO: Metabolically unhealthy obesity group.

**Table 1 nutrients-14-05223-t001:** Characteristics of studies evaluating the relationships of metabolic health status and obesity status with the risk of heart failure.

Study	Design (Duration)	Country	Sample Size	Definition of Metabolic Health	Obesity Assessment	Outcome Assessment (Cases)	Study Quality
Voulgari, 2011 [14]	Prospective (6.0 years)	Greece	550 men and women	Without metabolic syndrome	Overweight: BMI of 25–29.9 kg/m^2^; obesity: BMI of ≥30 kg/m^2^	MR (185)	High
Morkedal, 2014 [15]	Prospective (12.2 years)	Norway	61,299 men and women	Without metabolic syndrome	Overweight: BMI of 25–29.9 kg/m^2^; obesity: BMI of ≥30 kg/m^2^	MR (2547)	High
Caleyachetty, 2017 [18]	Prospective (5.4 years)	United Kingdom	3,495,777 men and women	Having <3 of diabetes, hypertension, or hyperlipidemia	Overweight: BMI of 25–29.9 kg/m^2^; obesity: BMI of ≥30 kg/m^2^	MR (25,254)	High
Cordola, 2021 [23]	Prospective (11.3 years)	United States	19,412 postmenopausal women	Having <2 of diabetes, hypertension, or hyperlipidemia	Overweight/obesity: BMI ≥ 25 kg/m^2^ or WC ≥ 88 cm	MR (455)	High
Itoh, 2021 [19]	Retrospective (3.1 years)	Japan	802,288 men and women	Without diabetes, hypertension, or hyperlipidemia	Overweight/obesity: BMI of ≥25 kg/m^2^	MR (588)	High
Kim, 2021 [20]	Prospective (6.0 years)	Korea	356,258 men and women	Having <2 of diabetes, hypertension, or hyperlipidemia	Overweight/obesity: BMI of ≥25 kg/m^2^	MR (5406)	High
Commodore-Mensah, 2021 [30]	Prospective (17.0 years)	United States	9477 men and women	Having <2 of diabetes, hypertension, or hyperlipidemia	Overweight: BMI of 25–29.9 kg/m^2^; obesity: BMI of ≥30 kg/m^2^	MR (1531)	High
Zhou, 2021 [16]	Prospective (11.2 years)	United Kingdom	381,363 men and women	Having <3 of diabetes, hypertension, hyperlipidemia, or C-reactive protein	Obesity: BMI of ≥30 kg/m^2^	MR (6215)	High
Bi, 2021 [31]	Prospective (9.7 years)	China	93,288 men and women	Having <2 of diabetes, hypertension, or hyperlipidemia	Overweight: BMI of 24–28 kg/m^2^; obesity: BMI of ≥28 kg/m^2^ or WC ≥ 80 cm	MR (1628)	High
Fauchier, 2021 [22]	Prospective (4.9 years)	France	2,873,039 men and women	Without metabolic syndrome	ICD-10 code E65	MR (391,637)	High

BMI: body mass index, MR: medical record, WC: waist circumference.

**Table 2 nutrients-14-05223-t002:** Subgroup analysis according to study location.

		Western Countries				Eastern Countries		
	No. of Studies	RR (95% CI)	*I^2^*	*p* for Heterogeneity	No. of Studies	RR (95% CI)	*I^2^*	*p* for Heterogeneity
MHOW	5	1.23 (1.17, 1.30)	39.4	0.16	3	1.23 (1.08, 1.39)	61.1	0.08
MHO	8	1.48 (1.38, 1.52)	87.9	<0.001	1	1.78 (1.45, 2.19)	-	-
MUHNW	8	1.65 (1.38, 1.98)	97.2	<0.001	2	1.35 (1.18, 1.55)	52.8	0.15
MUHOW	5	1.80 (1.52, 2.13)	90.6	<0.001	2	1.65 (1.54, 1.77)	-	-
MUHO	8	2.24 (1.50, 2.64)	97.5	<0.001	1	2.61 (2.22, 3.06)	-	-

MHOW: Metabolically healthy overweight group, MHO: Metabolically healthy obesity group, MUHNW: Metabolically unhealthy normal-weight group, MUHOW: Metabolically unhealthy overweight group, MUHO: Metabolically unhealthy obesity group.

## Data Availability

The data presented in the study are available in the included articles.

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
