# Peer review of "Risk of Heart Failure between Different Metabolic States of Health and Weight: A Meta-Analysis of Cohort Studies"

_nutrients, 2022, doi:10.3390/nu14245223_

Round 1
Reviewer 1 Report
The subject is of great interest and in general the research is well planned, executed and reported. However, there are some aspects that could be improved.
1. Title: the title should be improved to better reflect the topic investigated. It is proposed: "Risk of heart failure between different metabolic states of health and weight: a meta-analysis of of cohort studies".
2. Abstract: according to the PRISMA 2020 for Abstracts Checklist, the eligibility criteria should be provided in the abstract. It is also important that readers can identify the databases consulted by reading the abstract. It is also important to specify the methods used to assess risk of bias in the included studies.
3. Introduction: this section is adequately reported, it provides the reader with enough data to understand the problem and the objective of the review.
4. Methods - Literature search: In this section, despite having offered the keywords used, a better search strategy report is required in at least one database (present the full search strategy for almost one database).
5. Methods - Study selection: elegibility criteria are clear.
6. Methods - selection process: this section is missing [partially reported in another section]. Must specify the methods used to decide whether a study met the inclusion criteria of the review, including how many reviewers screened each record and each report retrieved, whether they worked independently, and if applicable, details of automation tools or software used in the process.
7. Methods - Data extraction: some information of this section is missing [partially reported in another section]. How many reviewers extracted data from each report, they used a piloted form, whether they worked independently, any processes for obtaining or confirming data from study investigators, and if applicable, details of automation tools or software used in the process.
8. Quality assessment: this section has been properly reported. The sentence: "Two authors (X.W and J.Y.D) in- 89 dependently performed the literature search, study selection, data extraction, and quality 90 assessment. Any disagreement was solved by discussion." must be placed in the correspondents sections.
9. Methods - Statistical analysis: this section has been properly reported.
10. Results - Study selection and characteristics: this section has been properly reported.
11. Results - Risk of heart failure: results about Risk of heart failure are well reported. However, given the observed heterogeneity, in addition to serving to choose the model, some subgroup analysis should have been carried out (within MHO, MUHNW, MUHOW, MUHO), to see if any characteristic of these studies could explain said heterogeneity (in addition to of the region, which partly explained the heterogeneity). It would be of interest to try to explain the heterogeneity.
12. Results - Risk of bias in studies: This information is only partially reported in Table 1. The authors must report, in addition to stating that they are high-quality studies, the score of each study, and if there was any aspect that suggests a risk of bias, detail it.
12. Discussion and conclusions: this section has been properly reported.
Reviewer 2 Report
Dear authors,
Congratulations for professionally implemented analysis of cohort studies related to heart failure risk.
As you mentioned, there are some limitations or weakness of this study. Despite that total number of observed participants is significant, the average value of the summary relative risks cannot reflect individual specific of participants with metabolic healthy overweight and metabolic healthy obesity in prediction of heart failure.
I would like to suggest you to continue this investigations taking into account the effect of sex, age and life style on increase/decrease of relative risk of heart failure for metabolic healthy people with overweight and obesity.
Author Response
Thanks for your comments. As we explained, we did not conduct sex-specific analyses because sex-specific asscoictions were not reported in most primary studies (Line 279-286). Similarly, other factors, such as age, socioeconomic status, or different lifestyles could also explain the heterogeneity. Unfortunately, detailed subgroup analyses according to these factors were not possible due to a lack of group-specific data. As you suggested, it is of great importance to conduct future studies to explore this association for individuals with certain characteristics. Text has been revised accordingly (Line 286-289).